# Lethal microbial blooms delayed freshwater ecosystem recovery following the end-Permian extinction

Chris Mays [1✉], Stephen McLoughlin [1], Tracy D. Frank[2], Christopher R. Fielding [2], Sam M. Slater[1] & Vivi Vajda [1✉]

Harmful algal and bacterial blooms linked to deforestation, soil loss and global warming are increasingly frequent in lakes and rivers. We demonstrate that climate changes and deforestation can drive recurrent microbial blooms, inhibiting the recovery of freshwater ecosystems for hundreds of millennia. From the stratigraphic successions of the Sydney Basin, Australia, our fossil, sedimentary and geochemical data reveal bloom events following forest ecosystem collapse during the most severe mass extinction in Earth's history, the end-Permian event (EPE; c. 252.2 Ma). Microbial communities proliferated in lowland fresh and brackish waterbodies, with algal concentrations typical of modern blooms. These initiated before any trace of post-extinction recovery vegetation but recurred episodically for >100 kyrs. During the following 3 Myrs, algae and bacteria thrived within short-lived, poorly-oxygenated, and likely toxic lakes and rivers. Comparisons to global deep-time records indicate that microbial blooms are persistent freshwater ecological stressors during warming-driven extinction events.

[1] Department of Palaeobiology, Swedish Museum of Natural History, Box 50007, SE-104 05 Stockholm, Sweden. [2] Department of Earth & Atmospheric Sciences, University of Nebraska-Lincoln, 126 Bessey Hall, Lincoln, NE 68588-0340, USA. ✉email: chris.mays@nrm.se; vivi.vajda@nrm.se

Algae and photosynthetic bacteria form the foundations of aquatic food webs, but their unconstrained proliferation can be lethal to animals. High abundances promote poorly oxygenated waters upon their death and decomposition, deplete dissolved oxygen via respiration during dark intervals, and many have toxic metabolic by-products[1,2]. Bloom events of microscopic algae (microalgae) and bacteria are triggered by high temperatures[1–3] and nutrient influx (e.g., from soil erosion following deforestation)[1,3]. Since they are exacerbated by increasing $CO_2$ and temperature, harmful microbial blooms are projected to become increasingly common into the future[2,4].

These environmental conditions were prevalent during the end-Permian event (EPE), the largest mass extinction in Earth's history[5]. The EPE has been linked to a marked increase in global temperatures (increase of c. 6–12 °C in the tropics[6], c. 10–14 °C at high southern latitudes[7]), and a rapid elevation of atmospheric $CO_2$ (c. 600% in <75 Kyrs)[8], caused by massive magmatic outgassing from the Siberian Traps Large Igneous Province[9]. The most consequential long-term changes on land included the abrupt demise of wetland glossopterid forests of the temperate Southern Hemisphere[10] and the tropical coal-forming forests of east Asia[11]. These were some of the most enduring and widespread biomes in Earth history, and their disappearance initiated a global "coal gap" in the rock record, reflecting a major reduction in atmospheric carbon drawdown that persisted for several million years[12]. Little is known of the subsequent continental recovery biomes for several reasons, including low ecosystem biomass, extensive oxidative weathering, and a scarcity of well-exposed, fossiliferous, non-marine strata of this age.

Long-term quantitative Permian and Triassic microbial records and algal concentrations from the continents are presently lacking, despite lakes and rivers being particularly susceptible to toxic microbial blooms, owing to their low turbulence, proximity to terrestrial nutrient sources, and long water residence times (e.g., in lowland lakes)[1]. A series of recent, high-precision age constraints[13–15], coupled with a well-resolved spore-pollen zonation scheme[10,16], now crowns the Sydney Basin, Australia, as the standard reference succession for upper Permian and Lower Triassic continental fossil and rock unit correlations in the Southern Hemisphere. These strata provide a near-continuous record of coastal plain environments through the EPE[13,14,16,17] (c. 252.2 Ma[14]), and span >4 Myrs of the post-EPE recovery interval[10,15] (Fig. 1). Here we present long-term, quantitative organic microfossil, sedimentary, and geochemical records of the Permian-Triassic transition from five stratigraphic sections in the Sydney Basin (Fig. 2 and Supplementary Data 1–16, Supplementary Figs. 1–6). The sections are tied to the regional biostratigraphic scheme[10] (see "Methods" section) and the results are integrated with previous palynological studies across eastern Gondwana. These successions reveal that the lowlands of eastern Gondwana provided long-term refugia for thriving algal and bacterial communities after the collapse of late Permian plant biotas. We propose that the proliferation of microbial communities was both a symptom of continental ecosystem collapse, and a cause of its delayed recovery.

## Results

The organic microfossil assemblages ("palynofacies"; Supplementary Data 1–4) define four successive ecological phases (see "Methods" section). Non-metric multidimensional scaling (nMDS) of the palynofacies supports these phases as indicated by their discrete regions in ordination space (Fig. 3).

### Pre-EPE (pre-252.2 Ma): Late Permian wetland communities and ecosystem collapse.
The "pre-EPE" phase is dominated by abundant wood, leaves, and pollen typical of wetland glossopterid gymnosperms (Fig. 2). Various other seed plants and diverse understorey ferns, sphenopsids, lycopsids, and bryophytes are also present, but are relatively less abundant. Algal diversity is relatively high, incorporating various chlorophyte green algae (e.g., *Cymatiosphaera*, leiosphaerids) and charophyte conjugating green algae (Zygnematophyceae; e.g., *Ovoidites*, *Peltacystia*, *Tetraporina*), but algal concentration values ($C_a$) are low (mean: ≤300 fossils/g), and abundances of other microbial remains (represented by amorphous organic matter; AOM) are very low (mean: c. 5%; Fig. 2). The $C_{org}/N_{total}$ is consistently high (Bootleg-8: mean = 15.73, s.d. = 9.5; Bunnerong-1: mean = 14.9, s.d. = 8.4), reflecting abundant organic remains derived from $C_3$ photosynthesis, the dominant metabolic pathway of land plants[18]. Pre-EPE coals and shales are typical of mires or well-vegetated river and coastal floodbasins[14] and contain abundant trace fossils of grazers and other primary consumers. Collectively, the fossil and geochemical data indicate that highly productive forest-mire ecosystems dominated the humid coastal plains of the Sydney Basin prior to the EPE (Fig. 4).

The widespread collapse of the late Permian southern high-latitude land and freshwater biomes (the glossopterid flora[10] and the *Peltacystia* Microalgal Province[19], respectively), occurred at c. 252.2 Ma[13,14] driven largely by warming and increased seasonality in precipitation[13] (Fig. 4).

### Early post-EPE (252.2–251.5 Ma): The microbial rising from the "dead zone".
In all successions examined, the ensuing "early post-EPE" interval (Fig. 2) initiates with sedimentary rocks typical of shallow standing water, and depauperate fossil assemblages of fungi, charcoal and other opaque wood fragments. This corresponds to the widely reported "dead zone" (*sensu* refs. [16,20], Fig. 2), an interval spanning several millennia during which widespread wildfires and deforestation[11,16] led to floodbasin water-table rise and ponding[14,16].

Our high-resolution analyses of post-EPE outcrop exposures reveal successive green algal associations as the first colonizers of continental environments following this "dead zone". The Frazer Beach succession expresses a major pulse of algal proliferation within the 12 m sampled interval (Fig. 2). This algal pulse initiated with a monotypic assemblage of smooth-walled "leiosphaerids" (*Leiosphaeridia*), a group that likely represents chlorophyte algae in the Sydney Basin (see Discussion)[19]. This was followed by two peaks of algal concentrations (61,000 and 70,000 fossils/g), which were dominated by zygnematophycean charophytes (*Circulisporites* and *Ovoidites*, both groups with >20,000 fossils/g each), but with leiosphaerid algae abundances remaining high (>10,000 fossils/g). Low concentrations of other chlorophyte groups (*Pediastrum*, *Quadrisporites*) were also recorded. Throughout this interval (0–160 cm), the Frazer Beach succession was deposited in very low salinity conditions (mean Sr/Ba mean = 0.06, s.d. = 0.05). The algal pulse terminates abruptly 160 cm above the EPE extinction horizon, replaced by high spore and pollen abundances through the succeeding 20 cm, accompanying the first occurrence of identifiable leaf fossils[16] (Fig. 2). Pollen typical of pre-EPE wetland floras ("glossopterid-type" herein) persist in low abundances for >4 million years after the dead zone; however, no post-EPE fossils can be definitively attributed to *Glossopteris* have yet been recovered from eastern Australia. Such pollen likely derives from alternative surviving groups of seed plants[10] and/or have been reworked from underlying strata.

The bore-core successions show that, on a broader stratigraphic scale, recurrent algal blooms occurred across the entire basin throughout the first c. 100 kyrs following the EPE (Fig. 2).

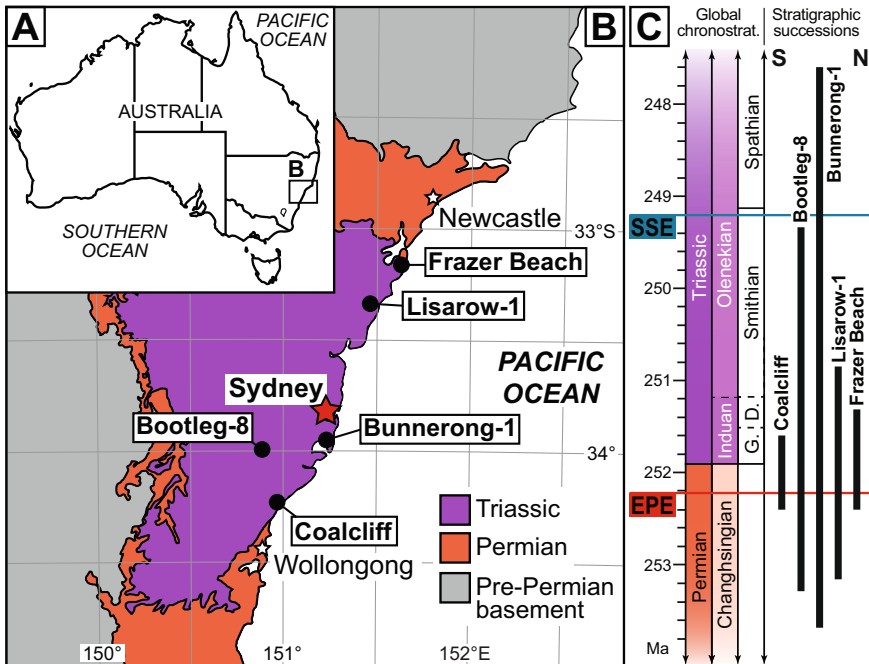

**Fig. 1 Geologic map and stratigraphy of the Sydney Basin, Australia. A** Map of modern Australia. **B** Geological map of southeastern Sydney Basin, with the locations of the studied core and outcrop successions. **C** Stratigraphic ranges of the examined Sydney Basin successions[10,13,16] and correlation to the global upper Permian and Lower Triassic chronostratigraphy (chronostrat.)[13–15]; EPE = end-Permian extinction event, Ma = millions of years ago, SSE = Smithian-Spathian climatic event, D. = Dienerian, G. = Griesbachian Stage, N = north, S = south.

In Bootleg-8 and Bunnerong-1, depositional conditions were fresh to brackish (Sr/Ba: Bootleg-8 mean = 0.11, s.d. = 0.05; Bunnerong-1 mean = 0.4, s.d. = 0.15) and algal assemblages were composed almost entirely of chlorophytes (leiosphaerids, *Quadrisporites*) at very high concentrations (Bootleg-8: 20,000–60,000 fossils/g). These values are consistent with blooms of chlorophyte algae within Quaternary freshwater/brackish lake sediments following local deforestation and/or enhanced nutrient influx[21,22].

Coincident with the algal blooms, the early post-EPE interval of the Sydney Basin also featured abundance maxima of amorphous organic matter (AOM; Fig. 2). Modern photosynthetic bacteria, such as cyanobacteria, produce the same granular texture, filamentous structure, and strong fluorescence[23,24] as the majority of AOM within this interval (Supplementary Fig. 4), indicating greatly enhanced rates of photosynthetic microbial productivity[23]. AOM abundances increased from pre-EPE levels of 2 or 11% to 33 or 41% during the early post-EPE phase (mean values from Bootleg-8 and Lisarow-1, respectively). The post-EPE AOM abundances are extraordinarily high for oxygenated continental depositional environments of fresh- to brackish salinity[25] (Fig. 2). Since organic matter derived from freshwater algae and bacteria has relatively low $C_{org}/N_{total}$ values[18], the severe reduction in $C_{org}/N_{total}$ values between the pre-EPE (Bootleg-8 and Bunnerong-1 combined mean = 15.3, s.d. = 8.9) to early post-EPE (mean = 7.0, s.d. = 5.5) intervals is consistent with elevated relative contributions by these microbial populations (Fig. 2). This relationship was supported herein by the negative correlations between $C_{org}/N_{total}$ and the abundances of AOM ($r = -0.4160$, $p = 0.0345$, $N = 26$) and algae ($r = -0.7258$, $p = 0.00003$, $N = 26$).

The early post-EPE phase saw greatly reduced evapotranspiration following the loss of wetland forests, with elevated water tables and inundation of the lowland floodplains by fresh- to brackish waters. Enhanced weathering intensity[7,13] and destabilization of soils following deforestation[16,26], promoted nutrient influx into the floodbasins. Combined with elevated $CO_2$[6],

temperature[6,7], and precipitation seasonality[7,13], these factors promoted numerous intermittent pulses of algal and bacterial proliferation.

**Late post-EPE (251.5–249.2 Ma): a recurrent microbial haven in the Early Triassic lowlands.** Throughout this >2.2 Myr-interval[10,13,14], bacterial and algal abundances were generally high until at least the late Smithian (Fig. 2). Combined mean AOM from Bootleg-8 and Lisarow-1 was c. 20%, and chlorophyte algal concentrations in Bootleg-8 were 5000–20,000 fossils/g. High algal and bacterial abundances are linked to consistently low $C_{org}/N_{total}$ values during the late post-EPE interval (Fig. 2), and algal (especially zygnematophycean charophyte) assemblages are of higher diversity compared to the previous phase. Sedimentary rocks of the coastal plain floodbasins contain abundant mudcracks and sparse trace fossils of benthic grazers (Fig. 4 and Supplementary Fig. 5). Sr/Ba ratios indicate consistently brackish or freshwater conditions in these temporary waterbodies.

Global $CO_2$ and temperatures were high throughout most of this interval[6–8], sustained by continued Siberian Traps magmatism until at least 250.60 Ma (Smithian)[27]. During this time, eastern Gondwana (c. 65–75°S) experienced strongly seasonal precipitation[7,13], and regular drying prevented the establishment of permanent wetland floras. delaying the return of high-latitude peat-mire carbon sinks until the Middle Triassic, prolonging the elevated global $CO_2$ levels. The floras of this "coal gap"[12] are represented by sparse leaf beds[16] and soils with weakly developed roots and isoetalean rhizomorph horizons[28], indicative of open, "dryland" (sclerophyll) forests of conifers, seed-ferns (Peltaspermales and Umkomasiales) and lycopsids[10]. Compared to pre-EPE wetland floras, the open post-EPE vegetation would have had relatively low biomass and evapotranspiration rates[29], facilitating seasonally high water tables and maximizing light availability to aquatic bacteria and algae[30]. Our combined data indicate that Dienerian–Smithian conditions were conducive to enduring

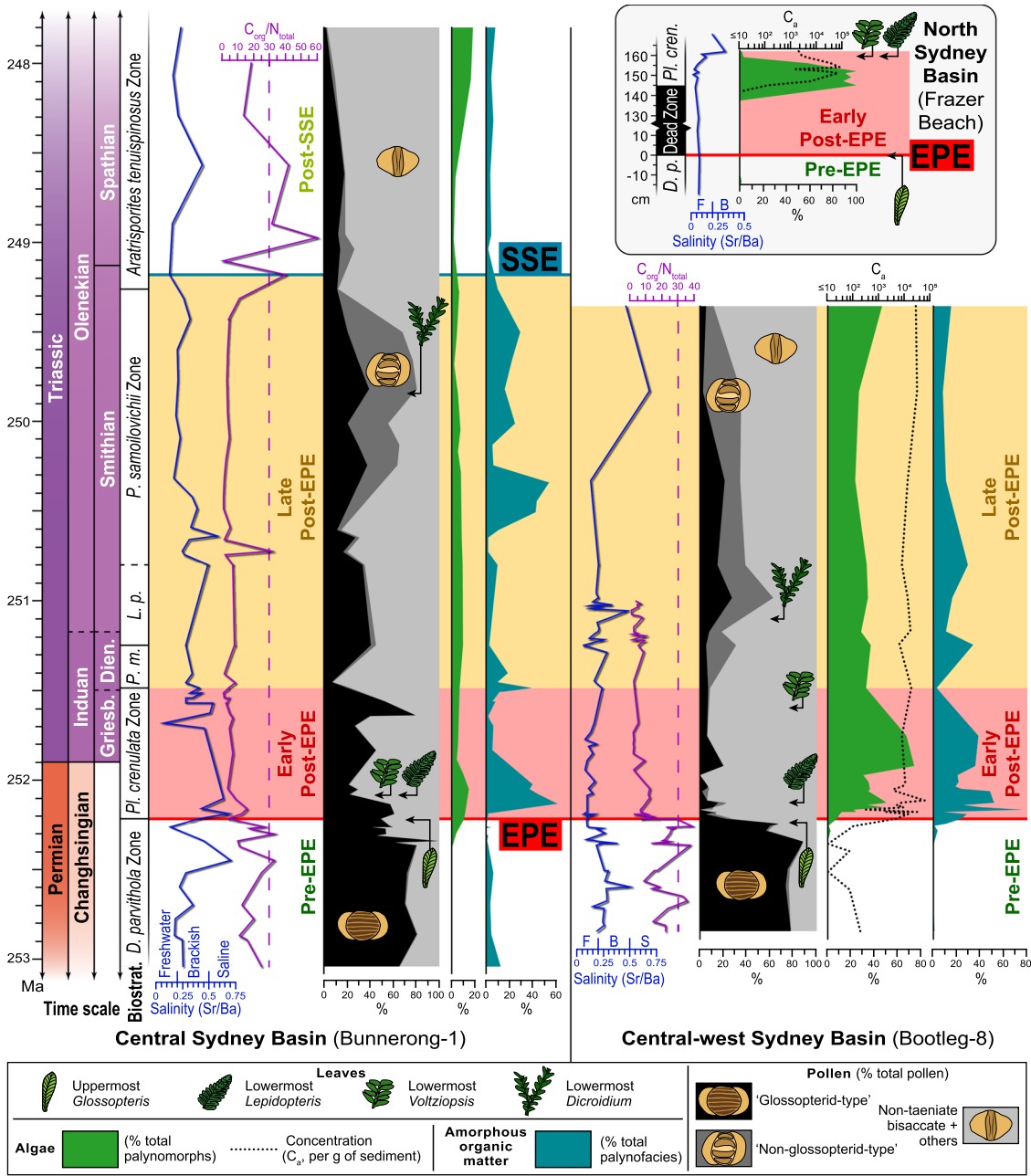

**Fig. 2 Microfossil and geochemical trends from the upper Permian to Lower Triassic of the Sydney Basin, Australia.** Data from Bootleg-8 and Bunnerong-1 are rescaled to the global chronostratigraphic scheme by anchoring spore-pollen biostratigraphic zone (biostrat.) boundaries to high-precision radiogenic isotope ages within the eastern Australian succession[13–15] as calibrated in refs. [10,14]. Bunnerong-1 amorphous organic matter and algal data from ref. [13]. Salinity category values follow ref. [82]. B brackish, F freshwater, S saline. Dashed lines at $C_{org}/N_{total} = 30$ for ease of comparison between successions. Note: glossopterid-type pollen were produced by both glossopterids and some other seed plants, "last *Glossopteris*" refers to leaf remains, not "glossopterid-type" pollen. "Dead zone" from ref. [16], EPE = end-Permian extinction event, Ma = millions of years ago, SSE = Smithian-Spathian climatic event, *D.* = *Dulhuntyispora*, *L. p.* = *Lunatisporites pellucidus* Zone, *P. m.* = *Protohaploxypinus microcorpus* Zone, *Pl.* = *Playfordiaspora*. Source data are provided as a Source Data file.

fresh/brackish-water ecosystems with sustained high abundances of algae and bacteria within fluctuating coastal plain waterbodies.

**Post-SSE (249.2–? Ma): the end of the microbial regime.** The first substantial changes in lowland ecosystems following the EPE occurred during the Smithian-Spathian event (SSE; c. 249.2 Ma). Regionally, this event was an interval of enhanced chemical weathering[13] and ecological instability[10]. In Bunnerong-1, the SSE is concurrent with a marked reduction in algal and AOM abundances, while a general increase in $C_{org}/N_{total}$ values reflects

a change in abundances from microbe-derived organic remains to those of land plants (Fig. 2). This concurs with global vegetation trends, characterized by widespread recovery of gymnosperm floras[10,31], promoted by climatic cooling[6].

**Discussion**

**Harmful microbial blooms across the post-extinction lowlands.** Following the end-Permian extinction, high abundances of algae and bacteria were facilitated by recurrent, dysoxic, fresh to brackish waterbodies across the floodbasins for more than three

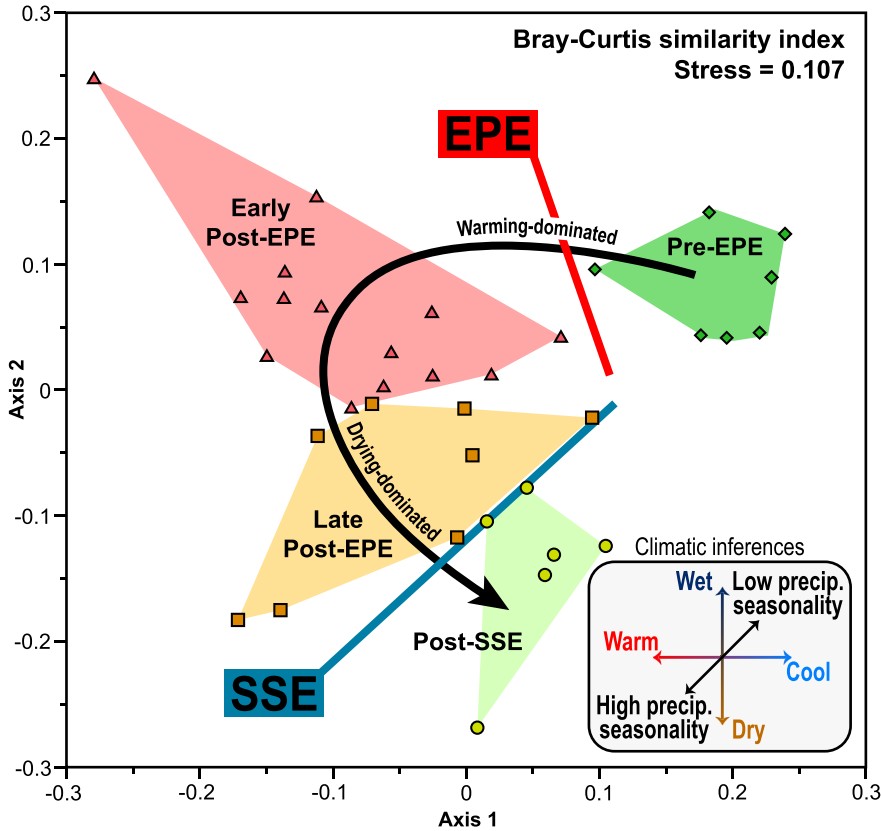

**Fig. 3 Non-metric multidimensional scaling (nMDS) plot for the ecological phases of the Late Permian to Early Triassic, based on organic microfossil assemblages of the Sydney Basin, Australia.** Insert box indicates inferred climatic changes through this interval, based on plant fossils, sedimentary features and climate modeling[7,12–14]. Large arrow indicates the progression of phases in stratigraphic order; EPE = end-Permian extinction event, precip precipitation, SSE = Smithian-Spathian climatic event. Source data are provided as a Source Data file.

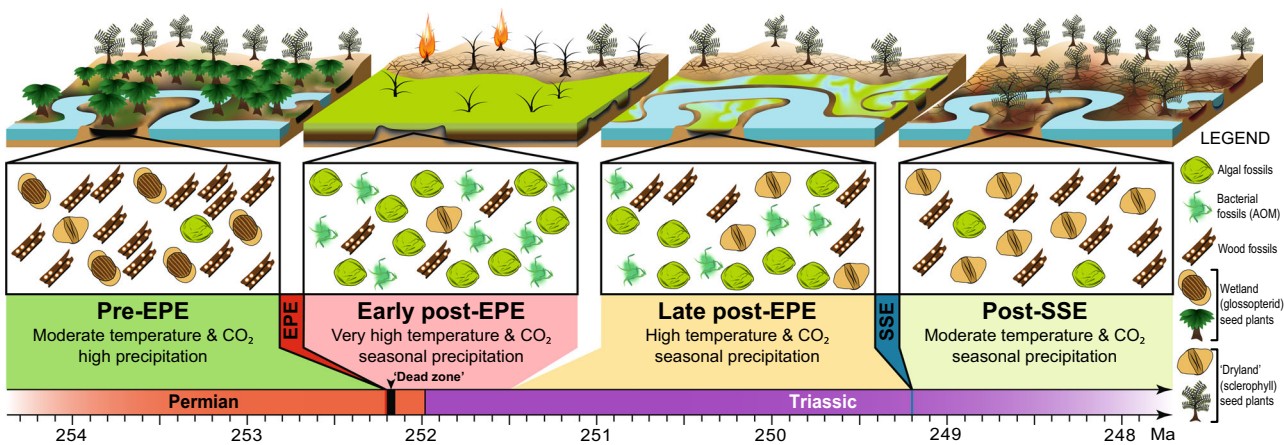

**Fig. 4 Reconstructions of late Permian to Early Triassic continental ecological phases with representative organic microfossil assemblages.** AOM = amorphous organic matter, EPE = end-Permian extinction event, Ma = millions of years ago, SSE = Smithian-Spathian climatic event.

million years (Figs. 2 and 4). For the early post-EPE, there is a strong negative correlation between algal abundance and salinity ($r = -0.6123$, $p = 0.00089$, $N = 26$), and the highest algal abundances are evident within the lowest salinity regions of the Sydney Basin (e.g., Bootleg-8, FBO; Fig. 2). Chemically stable polymers have been widely reported within the cell walls[32,33] of the nearest extant relatives of the Sydney Basin fossil algae[19]. Among extant algae, these polymers serve to prevent their desiccation while dormant in ephemeral waterbodies (e.g., lakes, ponds, streams) under seasonal climates[34,35]. Such conditions prevailed in the post-EPE lowlands across southern Gondwana[14] and would have

selectively promoted the proliferation of desiccation-resistant algae. The two most abundant algae (*Leiosphaeridia*, *Quadrisporites*) likely represent chlorophytes, based on morphological and architectural similarities to extant groups[19]. While *Leiosphaeridia* has a very long geological history and encompass groups of disparate relationships, the two alternative affinities for continental representatives of this genus from the Permian and Triassic (Trebouxiophyceae, prasinophytes[19]) are both chlorophytic[36]. In the early post-EPE interval, high abundances of other freshwater green algae (Zygnematophyceae: *Circulisporites*, *Ovoidites*) occurred within the area of lowest salinity (FBO;

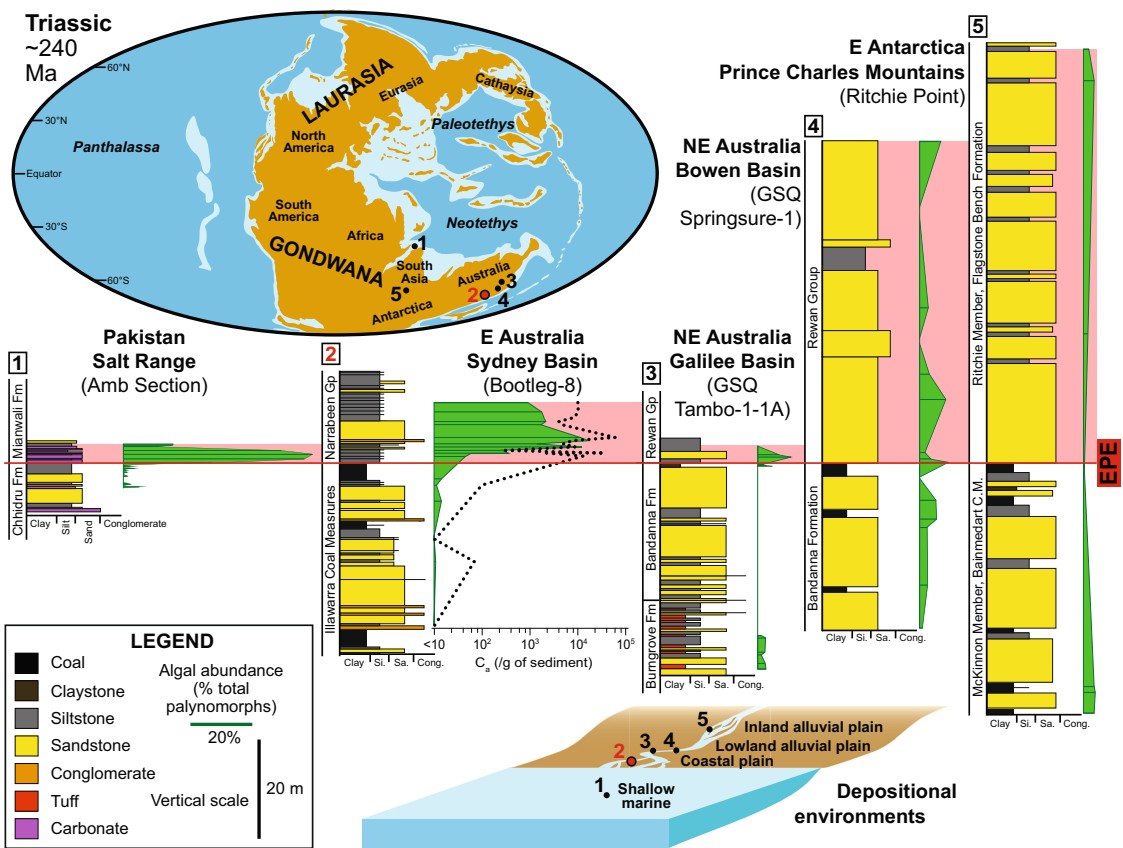

**Fig. 5 Gondwanan algae abundances across the end-Permian extinction event (EPE).** Triassic paleographic map indicates the Gondwanan localities compared here; map modified from ref. [84]. Relative abundances of algae are highest at, or soon after, the EPE at all Gondwanan sites. The exception is in the Prince Charles Mountains (PCMs) where sampling may be too sparse and sediment reworking has degraded the post-extinction microfossil signal. Relative abundances of algae are higher with decreasing depositional relief. Data sources (left to right): Salt Range[50], Sydney Basin (this study), Galilee Basin[37], Bowen Basin[44], PCMs[45]. $C_a$ = algal concentration, C.M. = Coal Measures, EPE = end-Permian extinction event, Ma = millions of years ago. Source data are provided as a Source Data file.

Fig. 2), but the proportion of non-chlorophytes is generally low across the Sydney Basin, and in continental deposits elsewhere[37]. High chlorophyte abundances in the post-EPE lowlands can be attributed to their tendency to proliferate in warm[3], nutrient-rich environments[38] and their competitive advantage under high $CO_2$ levels[39]. The dominance of this group over other groups (e.g., Zygnematophyceae) is likely due, in part, to their relative tolerance of brackish waters[19,40].

Owing to their propensity to decompose in oxygenated water[24,25], accumulations of cyanobacteria and other microbes were favored by low dissolved oxygen concentrations in the post-EPE floodbasins. This is supported by the scarcity of invertebrate animal trace fossils from this interval of the Sydney Basin[14] and other continental basins[41,42]. This also indicates the reduction of aquatic invertebrates through the EPE that would otherwise have prevented the preservation of abundant microbial remains via predation, detritivory, and sediment oxidation. Many of the conditions that promoted chlorophyte blooms in this interval (high $CO_2$, temperature, nutrient influx) similarly favor harmful cyanobacterial blooms in modern freshwater settings[1,2]. The optimal growth temperature range for both chlorophytes and harmful cyanobacteria in freshwater environments is c. 20–32 °C[1,3], which matches the estimated continental summer surface air temperatures (CSSATs) of eastern Gondwana for the earliest Triassic[7,13], and is within the range of projected mid-latitude CSSATs for the year 2100[43]. Bacterial and algal blooms in these intermittent waterbodies would have contributed to dysoxia and likely produced secondary metabolites toxic to animals[1],

impeding the recovery of Early Triassic freshwater ecosystems for at least 100 kyrs during this interval of recurrent microbial blooms.

In modern freshwater and marine environments, such blooms are considered a major threat to future ecosystem stability[2]. Our findings from the deep geological record underscore these warnings, and provide a disturbing prediction for the long-term consequences of continued warming and deforestation.

**Connecting and contrasting the end-Permian oceanic and continental records.** Comparison of the early post-EPE algal blooms of the Sydney Basin with other localities across the Gondwana supercontinent (Fig. 5) reveals concurrent relative abundance spikes of algae (and probable algal "acritarchs"; see "Methods" section). Continental successions spanning the EPE generally host higher relative abundances of fossil algae remains in more distal fluvio-lacustrine settings[37,44–46] (Fig. 5), highlighting improved opportunities for preservation in sluggish waterways (e.g., coastal plains). Higher energy fluvial settings, e.g., in the Permian-Triassic rift valley systems of central India[47] and East Antarctica[45], were less conducive to the preservation of extensive floodbasin deposits and accumulations of green algae, and more prone to sediment reworking, resulting in subdued algal abundances.

In marine settings, bacterial (including cyanobacterial) communities thrived in the aftermath of the EPE[48,49]. Similarly, marine algae experienced a near-global abundance "spike"[50–54] in strata immediately succeeding the EPE (Fig. 5); over the long-

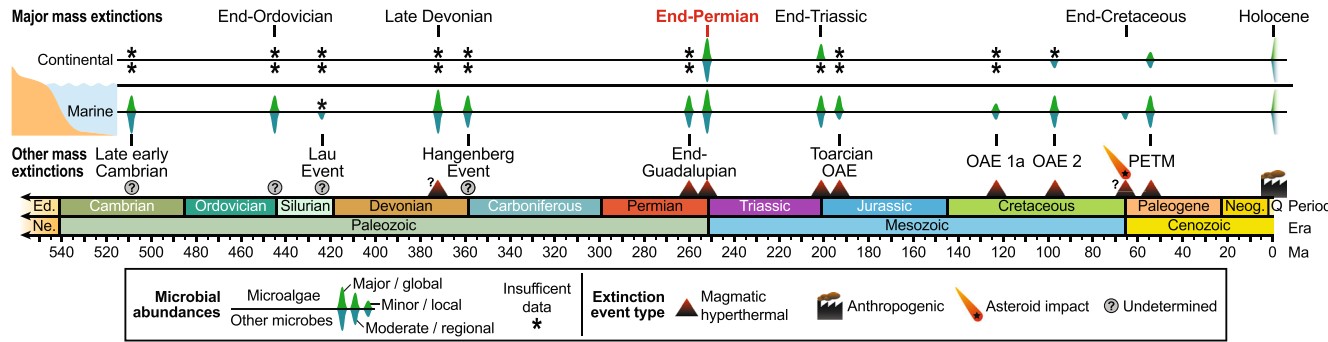

**Fig. 6 Phanerozoic mass extinctions with evidence of increased microbial abundances.** Inferred minor/local or moderate/regional microbial abundances may be in part due to inadequate data coverage. Extinction event types from ref. [59]. Data sources: late early Cambrian (c. 509 Ma)[63,85,86]; end-Ordovician (c. 445 Ma)[63,87]; Lau Event (c. 424 Ma)[64]; Late Devonian (c. 372 Ma)[62,63]; Hangenberg Event (c. 359 Ma)[62,63]; end-Guadalupian (c. 260 Ma)[19,88]; end-Permian (c. 252 Ma)[10,16,37,44,47,49-54,63,89]; end-Triassic (c. 183 Ma)[60,64,90-92]; Toarcian oceanic anoxic event (c. 183 Ma)[93-95]; Cretaceous OAE 1a (c. 123 Ma)[96]; Cretaceous OAE 2 (c. 94 Ma)[96-102]; end-Cretaceous (c. 66 Ma)[62,63,69,70,103-105]; Paleocene-Eocene Thermal Maximum (c. 56 Ma)[61,106-109]; Holocene[1,2,110,111]. Ed. = Ediacaran, Ma = millions of years ago, Ne. = Neoproterozoic, Neog. = Neogene, OAE = oceanic anoxic event, PETM = Paleocene-Eocene thermal maximum, Q = Quaternary.

term, however, marine algal concentrations decreased[51,55], suggesting an overall reduction in primary productivity[56]. The initial post-EPE increase of algae in nearshore environments has been interpreted as an in situ bloom linked to decreased salinity due to enhanced input of freshwater[57] and/or soil nutrient[58]. However, the most common algae in several coeval shallow marine records[50,51,55] are similar to the primary constituents of the continental Sydney Basin algal assemblages (e.g., *Leiosphaeridia*) suggesting that algae from freshwater blooms were transported to the marine realm en masse. Without accounting for this, productivity will be overestimated if based solely on early post-EPE shallow marine algal fossils.

**Extinction events and anachronistic freshwater ecosystems**. Magmatically triggered "hyperthermal" climates, characterized by very rapid increases in temperature and atmospheric $CO_2$, have been causally linked to most mass extinctions (e.g., EPE, end-Triassic event [ETE]) and numerous other biotic turnover events (e.g., Paleocene-Eocene Thermal Maximum [PETM])[59]. Microbe-dominated fossil assemblages have been widely reported from the marine records of hyperthermal events, but only rarely from continental successions, largely owing to the scarcity of studies (Fig. 6). Where documented, the continental assemblages of other rapid warming events reveal remarkably similar algal and bacterial increases to those of the post-EPE interval of the Sydney Basin. These include anomalously high abundances of leiosphaerids[60,61], other freshwater algae (ETE[60]), and/or AOM (PETM[61]). However, these latter events express microbial increases of lower magnitudes and shorter durations than those reported herein, likely owing to the less severe climatic shifts and the relatively modest continental ecosystem changes. Microbial proliferation during post-extinction intervals was largely a result of the significant reduction in other organisms that would normally consume, outcompete, or otherwise suppress the microbial populations[62-64]. In turn, we propose that recurrent microbial proliferation events likely hindered freshwater ecosystem recovery for most extinction events of the last several hundred million years.

An exception to this pattern is the end-Cretaceous extinction event (c. 66 Ma), which eliminated numerous large vertebrate clades, including the non-avian dinosaurs. This event was concurrent with, or immediately preceded by, a major episode of magmatism (the Deccan Traps Large Igneous Province), and an interval of associated warming[65]. Although this magmatism

may have contributed to the biotic turnover, the majority of extinctions are attributable to a very large asteroid impact[66], which injected voluminous dust and sulfate aerosols into the atmosphere, reducing incident solar radiation[67] and severely limiting marine microbial productivity[68]. Once light levels returned to pre-extinction levels, microbes underwent a short-lived, global proliferation[69], especially proximal to the impact site[70]. However, compared to other extinction events, this pulse was relatively minor and brief[62], probably due to the modest long-term increase in atmospheric $CO_2$ and temperature[71]. The contrasting microfossil signals for hyperthermal and asteroid impact events likely indicate the importance of sustained elevated greenhouse gas levels for the promotion of recurrent, harmful algal and bacterial blooms.

Post-EPE organic microfossil assemblages of the Sydney Basin resemble some of the earliest known continental ecosystems on Earth. Organically preserved fossils from the lake deposits of the Nonesuch Formation, USA (c. 1083–1070 Ma), reveal thriving communities of photosynthetic (cyano)bacteria, and eukaryotic assemblages dominated by *Leiosphaeridia*[72]. Although the specific organisms are unlikely to be closely related (e.g., *Leiosphaeridia* is polyphyletic[19]), the Sydney Basin post-EPE pulse of similar algae and cyanobacteria seems starkly anachronistic. "Anachronistic facies", typically represented by microbial communities preserved as inorganic sedimentary structures (microbialites)[63], have long been known to characterize post-EPE marine[73], and some continental, strata[41]. However, the present assemblages represent the first recorded organically preserved anachronistic communities in continental waterways following the EPE.

Continental ecosystem collapse paved the way for flourishing freshwater algal and bacterial communities in the wake of the largest mass extinction in Earth history: the end-Permian event (c. 252.2 Ma). Within the following c. 100 kyrs, prolific bacterial remains, and several pulses of high algal concentrations are evident in continental strata of the Sydney Basin, and in coeval lowland and shallow marine deposits across the Gondwanan supercontinent. Microbial proliferation was caused by the loss of lowland vegetation, which promoted high water tables, nutrient influx into waterways, and was intensified by extremely high $CO_2$ and temperature. Although a new flora appeared soon after ecosystem collapse, dense forest vegetation was absent for >3 Myrs, and lowland landscapes were regularly inundated by ephemeral, stagnant, fresh/brackish waterbodies hosting thriving algal and bacterial populations. Seasonal drought precluded the formation of peat-forming wetland mires, thus preventing the re-

establishment of these major carbon sinks during the Early Triassic, while microbe-derived toxins and dysoxia likely delayed the recovery of freshwater faunas. We highlight similarities of these microbial communities to some of the oldest known freshwater ecosystems, and postulate that such anachronistic continental fossil assemblages are symptomatic of major disruptions in freshwater ecosystems. Although continental records are sparse, Earth's global warming-driven extinction episodes are consistently linked to the proliferation of freshwater microbes, indicating that this is a recurrent phenomenon, and represents a disconcerting signal for future environmental change. These findings demonstrate the enduring deleterious impacts on continental biotas of climate-driven deforestation and prolonged high atmospheric greenhouse gas concentrations.

## Methods

All data derived from stratigraphic successions within the Sydney Basin, Australia (Fig. 1); new data were collected from 1, Australian Gas Light Company Bootleg DDH 8 (Bootleg-8); 2, Elecom Hawkesbury Lisarow DDH 1 (Lisarow-1); 3, Coalcliff outcrop (CCO); 4, Pacific Power Hawkesbury Bunnerong DDH 1 (Bunnerong-1); and 5, Frazer Beach outcrop (FBO; Supplementary Data 1–16).

**Palynology**. One hundred and fifty palynological samples were analyzed; 31 samples from Bootleg-8, 27 samples from Lisarow-1, 14 samples from CCO, 24 samples from Bunnerong-1, and 54 samples from FBO (Supplementary Data 1–11). Taxonomic categories are outlined in Supp. Data 12. Spore-pollen biostratigraphic zones and regional correlations follow ref. [10]. Palynological samples are housed at the Department of Palaeobiology, Swedish Museum of Natural History, Stockholm, Sweden.

To ensure the validity of inter-sample comparisons, all but two palynological samples (S029743, S029758) were collected from siltstone/claystone, heterolithic (interlaminated siltstone-fine sandstone), or coal lithofacies. These correspond to facies associations E, F, G, or H of ref. [14], interpreted as alluvial/coastal plain floodbasin (E, F), lake (G), or mire (H) palaeoenvironments. Palynological samples were digested using hydrochloric and hydrofluoric acids to remove inorganic mineral content. Quantitative analyses of samples from Bootleg-8, CCO, Lisarow-1, and FBO were undertaken via kerogen-based palynofacies abundance counts of ≥300 (FBO) or ≥500 (Bootleg-8, CCO, Lisarow-1) individual grains where possible (see Supplementary Data 1–4 for specimen categories). Bootleg-8, CCO, FBO, and Bunnerong-1 residues were oxidized with Schulze's Solution and sieved (Bootleg-8 and FBO: 5 μm; CCO and Bunnerong-1: 10 μm), followed by palynomorph counts of ≥100 specimens (FBO) or ≥250 (Bootleg-8, CCO, Bunnerong-1) where possible (Supplementary Data 5–11). Palynomorph concentrations and index taxon data were derived from slides that underwent this process (Supplementary Data 5–7). Light microscopy and photomicrography of fossil specimens were conducted using either a Zeiss Axioskop 2 Plus transmitted light microscope equipped with a Zeiss AxioCam MRc camera, or an Olympus BX51 transmitted light microscope equipped with a Lumenera Infinity 2 digital camera. AOM was categorized using standard light and fluorescence microscopy[24]. Fluorescence microscopy employed an Olympus BX51 equipped with an Olympus U-RFL-T fluorescence source (460–490 nm blue light excitation[74]) and an Olympus DP72 camera. All fluorescence photomicrographs were taken with 200 ms exposure lengths and ISO 200.

Concentration estimates were conducted using Lycopodium tablets from batches #3862 (Bootleg-8 and FBO [in part]), #140119321 (CCO), or #124961 (FBO [in part]), prepared by the Department of Geology, Lund University. Palynomorph concentration per gram of dried sediment ($C_p$) is derived by the following calculation (modified from ref. [75]):

$$C_p = (N \times t \times L_t)/(L \times M),$$

where $N$ = total palynomorph sample count, $t$ = number of Lycopodium tablets, $L_t$ = estimated Lycopodium spores per tablet ($L_t$ for Bootleg-8 and FBO [in part] = 9666, standard deviation [s.d.] = 671; $L_t$ for CCO = 19855, s.d. = 521; $L_t$ for FBO [in part] = 12542, s.d. = 931; see ref. [76]), $L$ = Lycopodium spores counted, and $M$ = mass of dried sediment. Algal concentration per gram of dried sediment ($C_a$) is calculated by:

$$C_a = (a/N) \times C_p,$$

where $a$ = total count of algae (+ acritarchs). Concentration estimates assume the minimal loss of palynomorphs or algal cysts during sample processing (including HF- and HCl-acid maceration, oxidation, and sieving) sample counts. Sample counts where no Lycopodium were encountered were excluded from estimates and illustrations since reliable concentration values could not be calculated. To standardize pre-burial biases, only samples from mudstone (claystone or siltstone) lithofacies were included in the illustrations.

**Phylogeny and terminology**. Algae (sensu lato) is an informal, polyphyletic group encompassing a broad range of photosynthetic eukaryotes, including evolutionarily disparate lineages such as Chromista (e.g., dinoflagellates) and the paraphyletic "green algae", which consists of all green plants (Viridiplantae) sans land plants[77]. Many of the fossils we report herein fall under the definition of "acritarch" outlined in ref. [78], which includes any small organic-walled fossil of unknown affinity. However, with the possible exception of Reduviasporonites[79], all of these have probable affinities to algae (s.l.)[19]; hence, we have generally included all such fossils under the term "algae" for concision. Unless specified, "algae" always refers to microscopic algae (or "microalgae"). Phylogeny of extant green algae follows ref. [36].

Microbe (adjective: microbial) fossils refer to the remains of any organism that can only be seen under a microscope; in the present study, this includes fossils of bacteria, microalgae, and acritarchs (N.B. excluding land plant remains, such as spores, pollen and wood/leaf fragments).

We use the term "palynofacies" in its originally defined sense[80] to refer to the total assemblage of hydrochloric- and hydrofluoric-acid insoluble organic remains of sediment or sedimentary rock sample.

**Ordination analysis**. Palynofacies assemblages were categorized by ecological phases, based primarily on the chemostratigraphy, biostratigraphy and palynofloral intervals of an earlier study[10]; these were (in stratigraphic order): 1, pre-EPE (=Dulhuntyispora parvithola Spore-pollen Zone); 2, early post-EPE (=Playfordiaspora crenulata Spore-pollen Zone, plus the "dead zone" sensu refs. [3,16], late post-EPE (=Protohaploxypinus microcorpus Spore-pollen Zone to the $\delta^{13}C_{org}$ excursion that marks the SSE[10,31], which corresponds to the lower A. tenuispinosus Spore-pollen Zone); and 4, post-SSE (=lower to mid-A. tenuispinosus Spore-pollen Zone). To test the validity of these time bins as discrete ecological phases, nMDS was employed on the palynofacies abundance data. Ordination analysis was conducted with the program PAST (v.4.03[81]).

To control for local variations in depositional conditions, all samples were from: 1, siltstone/claystone or heterolithic lithofacies; and 2, a single stratigraphic succession (Bootleg-8), except for the post-SSE population, since this interval only available from Bunnerong-1. A total of 35 assemblages met these selection criteria. The Bray–Curtis (dis)similarity index was derived for estimating the differences between palynofacies assemblages; prior to analysis, data were treated with square root transformations to constrain the palynomorph abundance ranges while preserving their rank-orders, and to facilitate direct comparisons with the previous analyses[10].

**Sedimentology**. Bore cores and surface exposures were logged sedimentologically, and successions were subjected to a facies analysis that is fully reported in ref. [14]. Facies associations were defined on the basis of lithology, nature of bed contacts, bed geometry (for outcrops), preserved physical and biogenic structures, color, associated fossils (including ichnofossils), and any other pertinent characteristics (see Table 1 of ref. [14]).

**Geochemistry**. The stable carbon isotope composition of bulk organic matter ($\delta^{13}C_{org}$) was determined on splits of samples used for palynological analysis in cores Bootleg-8 and Bunnerong-1. Total organic carbon ($C_{org}$), total nitrogen ($N_{total}$), and $\delta^{15}N$ values (not reported here) were measured simultaneously. In preparation for analysis, up to 500 mg of powdered sample was placed in a 50 ml centrifuge tube and reacted overnight with 1 N HCl at room temperature to remove carbonate mineral phases. Samples were then rinsed three times with ultra-pure water, with the supernatant separated by centrifugation (950×g for five minutes) and discarded. Samples were subsequently dried and crushed using an agate mortar and pestle. Samples were analyzed using a Costech Elemental Analyzer connected to a Thermo Finnigan MAT 253 stable-isotope gas-ratio mass spectrometer at the Keck-NSF Paleoenvironmental and Environmental Laboratory, University of Kansas, USA. Carbon isotope compositions are reported in per mil (‰) relative to the Vienna Peedee Belemnite (V-PDB) standard. Montana Soil (NIST Ref. Mat. 2711) and a calibrated yeast standard were used to monitor quality control, with compiled results over two years showing analytical error to be within ±0.22‰ for $\delta^{13}C_{org}$ values. Reproducibility of $C_{org}$ and $N_{total}$ analyses were monitored through replicate analyses of adenosine triphosphate (approximately one standard analysis for every 10 samples). Results indicate that reproducibility is better than ±0.09 and ±0.04 wt% for $C_{org}$ and $N_{total}$, respectively.

To aid in discriminating between fresh, brackish, and saline facies, the Sr/Ba paleosalinity proxy, proposed and tested in ref. [82], was applied to splits of $CaCO_3$-poor, mud-grade samples. In modern settings collected from a range of settings, these authors[82] distinguished between facies deposited within freshwater (Sr/Ba: <0.2), brackish (Sr/Ba: 0.2–0.5) and marine/saline (Sr/Ba: >0.5) environments, with a prediction accuracy of c. 66%. Sr and Ba concentrations were determined using a Bruker Tracer 5i portable X-ray fluorescence (XRF) analyzer in the Sedimentary Geochemistry Laboratory, University of Nebraska-Lincoln, USA. The instrument was calibrated with a series of mudrock standards that include a suite of reference materials characterized by ref. [83] and nine in-house reference materials from Permo-Triassic mudrocks in Bunnerong-1[13]. Ba concentrations were analyzed with no filter at 15 kV and 15 μA and a count time of 30 s, with one analysis measured under a helium flush to negate atmospheric interference. Sr concentrations were determined at 50 kV and 35 μA and a count time of 30 s, with a Cu 100 μm:Ti

25 μm:Al 200 μm filter. For all analyses, the raw count rate per second (rcps) signals lay between 20,000 and 100,000 rcps. Reproducibility of Sr and Ba analyses was monitored through replicate analysis of samples, at a rate of approximately one replicate analysis per 20 samples. Results indicate that reproducibility is ±5 ppm for Sr and ±124 ppm for Ba.

Statistical correlations between coeval palynological and geochemical samples employed the Pearson correlation coefficient (Pearson's $r$), and statistical significance $p$ values (two-tailed) were calculated for each. Full correlation statistics are provided in Supplementary Data 16.

**Reporting summary**. Further information on research design is available in the Nature Research Reporting Summary linked to this article.

## Data availability
All data generated or analyzed during this study are included in this published article (and its Supplementary Information files). Source data are provided with this paper.

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

## Acknowledgements

Bob Nicoll, Jim Crowley, and Malcolm Bocking for advice on radiogenic isotope age controls and stratigraphic relationships of Sydney Basin units. Cornelia & Arne Winguth for advice on Early Triassic climates. Jane Sibbons and Robert Hill at the University of Adelaide for assistance with fluorescence microscopy. This research was funded by the Swedish Research Council (grants 2015-04264, 2018-04527, 2019-04061, 2019-04524) and the National Science Foundation (grant EAR-1636625).

## Author contributions

C.M.: Conceptualization, data curation, palynological and palaeoenvironmental analysis, and writing the original draft. V.V.: Funding acquisition, conceptualization, data curation, palynological analysis, writing, and editing. S.M.: Funding acquisition, conceptualization, data curation, macrofloral analysis, writing, and editing. T.F.: Funding acquisition, data curation, geochemical analysis, writing, and editing. C.F.: Funding acquisition, data curation, sedimentological analysis, writing, and editing. S.S.: Comparative mass extinction dynamics and algal ecology, writing, and editing.

## Competing interests

The authors declare no competing interests.
