## [Peer Review File · Nature Communications]

Reviewers' Comments:

Reviewer #1:

Remarks to the Author:

The paper "Lethal microbial blooms delayed freshwater ecosystem recovery following the end-Permian extinction" by Mays et al. deals with a very important and interesting topic focused on a significant geological time interval. The paper is surely suitable for the journal being of broad international interest. Starting from the data obtained from the stratigraphic successions of the Sydney Basin, Australia, the authors document an extreme microbial bloom events following forest ecosystem collapse during the end-Permian event, one of the most severe mass extinction events in Earth's history. The conclusions are original and novel and quantitatively demonstrated. The subject dealt is matter of debate within the international scientific community and this excellent paper provides data and conclusion which surely will contribute to increase the knowledge. Generally speaking, the paper is well written and properly organized. All the illustrations and tables are necessary, of high quality and very clear. I have not a lot of comments except for suggesting a reference that should be useful to compare the data from the Sydney Basins with those of a key section bracketing the P/T boundary in the Southern Alps (Italy) (Spina et al., 2015 with references) where a proliferation of *Reduviasporonites chalastus*, recorded widespread occurring before and during the EPE, have been documented at different stratigraphic intervals. Spina et al. (2015 with references) interpreted *R. chalastus* as a chlorophycean alga resistant to processes such as bacterial degradation and chemical breakdown and with palaeoecological preference probably related to anomalous water salinity ranging.

The other references are adequate and all are necessary.

Spina, A. Cirilli, S., Utting, J., Jansonius, J. (2015). Palynology of the Permian and Triassic of the Tesero and Bulla sections (Western Dolomites, Italy) and consideration about the enigmatic species *Reduviasporonites chalastus*. *Review of Palaeobotany and Palynology* 218 (2015) 3–14, <http://dx.doi.org/10.1016/j.revpalbo.2014.10.003>

regards

Simonetta Cirilli

Reviewer #2:

Remarks to the Author:

The manuscript deals with Permian/Triassic freshwater-brackish, deltaic, settings of the Sydney Basin and reaches robust paleoenvironmental inferences on the basis of palynological, geochemical and sedimentological evidence. A key aspect of the study relates with microbial activity (presumed to be cyanobacterial) inferred from indirect palynological evidence (presence of AOM in palynofacies) and geochemical proxies (Corg/Ntot), the latter being, arguably, the most convincing of the two. The authors convincingly describe an extremely likely palaeoenvironmental scenario and the study is remarkable and does not include any flaws. For this reason, I recommend his publication with very minor corrections, essentially concerning the rephrasing of certain sentences and expressions (proposed below). The overall quality of the data is considered as being very high and the main findings are of high interest for a general audience.

On line 47, I would delete the 'long water residence times' from the sentence as residence time of continental water bodies is quite heterogeneous. In fact, this very aspect is quite important as planktonic communities and ecologic (primary) productivity are known to differ substantially in systems characterized by shorter (e.g., river) vs. longer residence time (e.g., reservoir-type, lake).

In Fig. 2, the use of the expression 'Glossopterid-type' is awkward and should be replaced by another one since the uppermost occurrences of glossopterid mega-fossil remains could be interpreted to be a Last Appearance Datum of the group. Maybe the authors could use "striate bisaccate type 1" and "Striate bisaccate type 2" or alternatively replace "last glossopterids" by "uppermost occurrence of glossopterid leaves" in Fig2 in order to get rid of the ambiguity.

Line 95 and following: The authors relate leiosphaerids to the green algae Chlorophyta. While some leiosphaerids may be referred to green algae, it would be extremely unlikely than the bulk of the fossil specimens presenting this extremely simple morphological characteristic be actual representatives of this group of algae. The authors are following the attribution given by Mays et al. (2021, *Earth-Science Review*, vol. 212, a paper representing probably the most comprehensive treatment of fossil algal palynomorphs since Tappan, 1980). In that paper, the authors (op. cit., p. 10, section 3.4.1.2) further indicated that Leiosphaeridia, a palynomorph with a "stratigraphic range extending at least as far back as the Late Palaeoproterozoic", is "probably polyphyletic and [that] any single affinity is unlikely to capture the diversity encompassed by this fossil." (e.g. not only from the phylum Chlorophyta) In other words, an attribution to green algae could be an over simplification and I would recommend a more conservative approach and a reference to 'acritarchs'. It has to be highlighted that the use of one name or the other does not invalidate any of the paleoenvironmental inferences and conclusions of the study.

Line 83: the authors used the word "palynofacies" to refer to the fossil database used to perform their multivariate analysis. This term has various acceptations. While the term palynofacies has been used to denominate organic particles of any types (e.g., phytoclasts, palynomorphs, AOM...) (sensu Combaz, 1964, *Rev Micropal*, vol. 7; or Tyson, 1995), Australian palynologists tend to include only palynomorphs (e.g. study of palynological assemblage). The meaning of this word should be indicated (in supplementary data) and the data used highlighted more precisely (supplementary data includes many tables).

Fig. 3. The figure is extremely well achieved but additional information should be given in the caption (or in supplementary data) concerning the small insert on the lower right (Cool-wet-warm-dry). In my opinion, the fig should not include references on climate as these paleoenvironmental(climatic) inferences, though probably valid, are done on the basis of indirect inferences? (Pre-EPE arborescent wetlands, climate modelling presented earlier?)

Line 96 and following: the authors use the word charophytes to refer to the fossil zygospores Ovoidites, Peltacystia and Tetraporina. While the use of this high taxonomic group is not scientifically incorrect, as Charophyta includes Zygnematophyceae, it may be misleading as the clade Charophyta includes many other algal groups and, in particular, the Coleochaetales and the stoneworts (Charophyceae). If the authors have not found any fossil remains attributed to the latter, I would recommend to use the taxonomic group Zygnematophyceae, which is more restricted.

lines 190-192: reference to modern groups with low salt-tolerance and toxicity are somewhat far-reaching and not needed.

line 198: I would delete the reference Number 33 as the group mentioned in the reference is not represented in the assemblages. The sentence between line 195 and line 198 could be slightly rephrased to delete any reference to Trebouxiophyceae.

lines 205-207: This sentence depend on the assumption that the leiosphaerids of the Sydney Basin are actually chlorophytes. I would tend to believe that the evidence for that could be questioned but I reckon that the resulting paleoecological scenario is extremely likely. The authors should also acknowledge that the changes in the proportion of leiosphaerids and Zygnematophyceae could be due to other paleoenvironmental factors.

line 208: misspelling replace oxic by sub-oxic

line 248: I believe that the authors refer to the productivity to shallow marine algae. It would be less ambiguous to replace 'shallow marine algal assemblages' by 'shallow marine algal productivity'.

Fig. 6 is extremely well-achieved but some expression should be slightly rewritten as algae are equated to microbes which is an oversimplification. Please replace 'Microbial abundance' by 'Microbial and algal abundance' and replace 'other microbes' by 'microbes'

The reference to algal (and prokaryotic, e.g., Siphonophycus spp.) assemblages from the 1Gyr-old Torridon Gp is welcome but the authors should also insert a reference to other Meso-/Neoproterozoic assemblages which are taxonomically better constrained (e.g. Roper Gp, Australia). It has to be highlighted that virtually any pre-Ediacaran palynological assemblage is dominated by or include large proportions of leiospheres.

Reviewer #3:

Remarks to the Author:

Researchers studying mass extinctions often focus on the taxa that go extinct, while the surviving taxa can actually provide information on what happens with ecosystems during the survival and recovery interval. And this is exactly what Mays et al. do, focusing on the unusual long-term proliferation of algae in the Sydney Basin, comparing these patterns with other areas in southeastern Gondwana, and similar signals during other mass extinction events.

P 2, line 35. "extreme global warming (c. 6–12°C increase in the tropics)". Just by looking at the current warming of the boreal regions, a potentially threefold increase in CO₂ levels must have raised the temperatures and changed the rainfall regime in the higher latitudes immensely. There have been other papers that modeled the temperature increase as a result of the CO₂ outgassing of the Siberian Traps; please expand the number of sources for climate reconstruction. When doing so, could the authors give us an idea of how gradual the temperature increase would have been, given that the traps have been active over quite a long period, and what could have caused the system during that long period of Traps emissions to reach a threshold (the EPE) and trigger sudden biotic change over a large geographic region? Also, what would have been the magnitude estimated temperature increase for the high latitude regions at the end of the Permian?

P 2, lines 36-38. "The most consequential long-term changes on land included the abrupt demise of wetland glossopterid forests of the temperate Southern Hemisphere 8, 9 and the tropical coal-forming forests of east Asia". The text here seems to indicate that members of the glossopterid lineage disappeared during the EPE and based on the terminating Glossopteris icon in Fig 2. and earlier papers by the same team that is the case. However, in the same figure 2, the black shaded areas seem to indicate pollen produced by glossopterids and the palynomorph category continues to be present until the very top of the figure (Spatian). If the black areas represent similar pollen types, but produced by different parent plants than glossopterids, that has to be clarified in the text and figure caps. If the Glossopteris pollen decline but linger on, could the authors then indicate when this group roughly disappears from the Sydney Basin?

The glossopterids, which formed dominant vegetation in southern Gondwana, disappear when the concentration of algal remains goes through the roof. These two events almost certainly were brought about by the same ultimate cause, the periodic eruption of the Siberian Traps. Based on conditions during extant algal bloom phenomena, the authors argue that climate change caused the decline of glossopteris, which lead to deforestation, soil erosion and eutrophication of fresh and shallow marine water bodies. This sounds like a plausible explanation for what happens at the EPE, but it is not clear to me how this would have inhibited the recovery of freshwater ecosystems for hundreds of millennia. Palynological records show that other groups (terrestrial plants) establish themselves rapidly in the lowlands, including the more drought and high temperature tolerant seed ferns *Dicroidium* and *Lepidopteris*, and at a later stage the conifer *Voltziopsis*. These taxa that must have been present in the region were clearly better adapted to the new conditions and seemed to have been permanent elements of a relatively stable low diversity recovery flora. Is there evidence that for soil erosion during the Late Post EPE, and were the Siberian Traps still active during this time? If not, what could have been the cause for the algal dominance during that phase?

Supplementary Figure 3 and 4. The botanical affinity of the pollen and spores seem in order, but it would be good to provide information on the sources on which they are based. That way the readers don't have to go looking for this information.

Reviewer comments (in black) and author responses (in blue)

Reviewer #1 (Remarks to the Author):

The paper “Lethal microbial blooms delayed freshwater ecosystem recovery following the end-Permian extinction” by Mays et al. deals with a very important and interesting topic focused on a significant geological time interval. The paper is surely suitable for the journal being of broad international interest. Starting from the data obtained from the stratigraphic successions of the Sydney Basin, Australia, the authors document an extreme microbial bloom events following forest ecosystem collapse during the end-Permian event, one of the most severe mass extinction events in Earth’s history. The conclusions are original and novel and quantitatively demonstrated. The subject dealt is matter of debate within the international scientific community and this excellent paper provides data and conclusion which surely will contribute to increase the knowledge.

Generally speaking, the paper is well written and properly organized. All the illustrations and tables are necessary, of high quality and very clear. I have not a lot of comments except for suggesting a reference that should be useful to compare the data from the Sydney Basins with those of a key section bracketing the P/T boundary in the Southern Alps (Italy) (Spina et al., 2015 with references) where a proliferation of *Reduviasporonites chalastus*, recorded widespread occurring before and during the EPE, have been documented at different stratigraphic intervals. Spina et al. (2015 with references) interpreted *R. chalastus* as a chlorophycean alga resistant to processes such as bacterial degradation and chemical breakdown and with palaeoecological preference probably related to anomalous water salinity ranging.

- We have now included reference to this work in the appropriate part of the text (manuscript and supplementary materials).
- We agree that this genus is particularly interesting, but due to space restrictions, we were unable to discuss it at great length in the present manuscript. We briefly mentioned *Reduviasporonites* in the Supplementary Materials, and direct readers to a recent paper in which CM, VV and SM reviewed the non-marine records of algae and probable algal acritarchs including *Reduviasporonites* (*Earth-Science Reviews*, 212: 103382).

The other references are adequate and all are necessary.

Spina, A. Cirilli, S., Utting, J., Jansonius, J. (2015). Palynology of the Permian and Triassic of the Tesero and Bulla sections (Western Dolomites, Italy) and consideration about the enigmatic species *Reduviasporonites chalastus*. *Review of Palaeobotany and Palynology* 218 (2015) 3–14, <http://dx.doi.org/10.1016/j.revpalbo.2014.10.003>

regards

Simonetta Cirilli

Reviewer #2 (Remarks to the Author):

The manuscript deals with Permian/Triassic freshwater-brackish, deltaic, settings of the Sidney Basin and reaches robust paleoenvironmental inferences on the basis of palynological, geochemical and sedimentological evidence. A key aspect of the study relates with microbial activity (presumed to be cyanobacterial) inferred from indirect palynological evidence (presence of AOM in palynofacies) and geochemical proxies (Corg/Ntot), the latter being, arguably, the most convincing of the two. The authors convincingly describe an extremely likely palaeoenvironmental scenario and the study is remarkable and does not include any flaws. For this reason, I recommend his publication with very minor corrections, essentially concerning the rephrasing of certain sentences and expressions (proposed below). The overall quality of the data is considered as being very high and the main findings are of high interest for a general audience.

On line 47, I would delete the 'long water residence times' from the sentence as residence time of continental water bodies is quite heterogeneous. In fact, this very aspect is quite important as planktonic communities and ecologic (primary) productivity are known to differ substantially in systems characterized by shorter (e.g., river) vs. longer residence time (e.g., reservoir-type, lake).

- Good suggestion; we have clarified the sentence to indicate that long residence times only apply to some continental waterbodies such as lakes.

In Fig. 2, the use of the expression 'Glossopterid-type' is awkward and should be replaced by another one since the uppermost occurrences of glossopterid mega-fossil remains could be interpreted to be a Last Appearance Datum of the group. Maybe the authors could use "striate bisaccate type 1" and "Striate bisaccate type 2" or alternatively replace "last glossopterids" by "uppermost occurrence of glossopterid leaves" in Fig2 in order to get rid of the ambiguity.

- To avoid the ambiguity, we have included a short explanation in the text and caption to clarify this, and re-labelled the leaves in Fig. 2 ('Glossopteris' rather than 'glossopterid').

Line 95 and following: The authors relate leiosphaerids to the green algae Chlorophyta. While some leiosphaerids may be referred to green algae, it would be extremely unlikely than the bulk of the fossil specimens presenting this extremely simple morphological characteristic be actual representatives of this group of algae. The authors are following the attribution given by Mays et al. (2021, Earth-Science Review, vol. 212, a paper representing probably the most comprehensive treatment of fossil algal palynomorphs since Tappan, 1980). In that paper, the authors (op. cit., p. 10, section 3.4.1.2) further indicated that Leiosphaeridia, a palynomorph with a "stratigraphic range extending at least as far back as the Late Palaeoproterozoic", is "probably polyphyletic and [that] any single affinity is unlikely to capture the diversity encompassed by this fossil." (e.g. not only from the phylum Chlorophyta) In other words, an attribution to green algae could be an over simplification and I would recommend a more conservative approach and a reference to 'acritarchs'. It has to be highlighted that the use of one name or the other does not invalidate any of the paleoenvironmental inferences and conclusions of the study.

- We would argue to maintain the chlorophyte designation for the Sydney Basin populations of *Leiosphaeridia*. In support of this, the reviewer cites the recent review by CM, VV & SM (*Earth-Science Reviews*, 212: 103382) wherein the affinities of the Sydney Basin leiosphaerids are concluded to be Trebouxiophyceae or prasinophytes; both of these groups

are chlorophytes (*sensu stricto*; Dal Corso *et al.* 2020, *PNAS* 117: 2551–2559, reference now included in supplementary materials). As such, we have chosen the taxonomic designation that is adequately broad to cover both. (Although, as outlined in the review paper, Trebouxiophyceae may be more likely for the Sydney Basin populations, given the preference for continental conditions among extant members of this group).

- However, we appreciate the suggestion that we should not invite the readers to interpret all occurrences of *Leiosphaeridia* as chlorophytes. We have now provided a clarification in the manuscript.
- ‘Acritarch’ has been utilized only where an affinity cannot be confidently assigned to any algal group (e.g., *Reduviasporonites*, *Michrhystridium*). A discussion of this term is included in the supplementary text.

Line 83: the authors used the word “palynofacies” to refer to the fossil database used to perform their multivariate analysis. This term has various acceptations. While the term palynofacies has been used to denominate organic particles of any types (e.g., phytoclasts, palynomorphs, AOM...) (*sensu* Combaz, 1964, *Rev Micropal*, vol. 7; or Tyson, 1995), Australian palynologists tend to include only palynomorphs (e.g. study of palynological assemblage). The meaning of this word should be indicated (in supplementary data) and the data used highlighted more precisely (supplementary data includes many tables).

- This is an excellent suggestion. A working definition of palynofacies has now been included to clarify our usage (we follow the former definition). The specific dataset used for the analyses is indicated in the appropriate position (Supp. Table 1).

Fig. 3. The figure is extremely well achieved but additional information should be given in the caption (or in supplementary data) concerning the small insert on the lower right (Cool-wet-warm-dry). In my opinion, the fig should not include references on climate as these paleoenvironmental(climatic) inferences, though probably valid, are done on the basis of indirect inferences? (Pre-EPE arborescent wetlands, climate modelling presented earlier?)

- We have now included a title for this insert that highlights that these are inferred climatic signals, and offer support for these inferences in the caption based on our previous palaeobotanical, sedimentological and palaeoclimatic analyses (references have been added).

Line 96 and following: the authors use the word charophytes to refer to the fossil zygospores Ovoidites, Peltacystia and Tetraporina. While the use of this high taxonomic group is not scientifically incorrect, as Charophyta includes Zygnematophyceae, it may be misleading as the clade Charophyta includes many other algal groups and, in particular, the Coleochaetales and the stoneworts (Charophyceae). If the authors have not found any fossil remains attributed to the latter, I would recommend to use the taxonomic group Zygnematophyceae, which is more restricted.

- We agree. This has been updated throughout the text.

lines 190-192: reference to modern groups with low salt-tolerance and toxicity are somewhat far-reaching and not needed.

- Removed.

line 198: I would delete the reference Number 33 as the group mentioned in the reference is not represented in the assemblages. The sentence between line 195 and line 198 could be slightly rephrased to delete any reference to Trebouxiophyceae.

- We would argue to retain the references in this sentence, since they highlight the function of these polymers in both Zygnematophyceae (Graham & Gray, 2001) and Trebouxiophyceae (Demura et al., 2014). These groups are the probable affinities of the most common algae within the Sydney Basin (see Mays et al. 2021, *Earth-Science Reviews*, 212: 103382).
- To qualify these points, we have made modifications to this section of text. Furthermore, we have added one reference that provides a modern perspective on chlorophyte wall composition, and emphasises that such desiccation-resistant polymers are present in many chlorophyte groups (including prasinophytes and Trebouxiophyceae).

lines 205-207: This sentence depend on the assumption that the leiosphaerids of the Sydney Basin are actually chlorophytes. I would tend to believe that the evidence for that could be questioned but I reckon that the resulting paleoecological scenario is extremely likely. The authors should also acknowledge that the changes in the proportion of leiosphaerids and Zygnematophyceae could be due to other paleoenvironmental factors.

- The point regarding the affinity of the leiosphaerids has been addressed above, and the manuscript text has now been updated accordingly.
- Regarding chlorophyte dominance over Zygnematophyceae: the text has been modified to reflect that the salinity differences only partly explain this, but leaving open the possibility of additional environmental factors.

line 208: misspelling replace oxic by sub-oxic

- In the context, high dissolved oxygen is intended (i.e., oxic rather suboxic); however, to avoid misinterpretations, 'oxic conditions' has been replaced by 'oxygenated water'

line 248: I believe that the authors refer to the productivity to shallow marine algae. It would be less ambiguous to replace 'shallow marine algal assemblages' by 'shallow marine algal productivity'.

- This sentence has been reconstructed to clarify this point.

Fig. 6 is extremely well-achieved but some expression should be slightly rewritten as algae are equated to microbes which is an oversimplification. Please replace 'Microbial abundance' by 'Microbial and algal abundance' and replace 'other microbes' by 'microbes'

- The usage of microbe in this paper refers to any microorganism; hence, the microalgae reported herein would fall under that definition. Hence, with this definition, the use of 'microbial and algal abundance' would be erroneous, since 'microbial' would encompass microalgae.
- However, we appreciate that the reviewer has highlighted a source of potential confusion. The figure and caption have been modified to better reflect that microbes include the microalgae reported herein.
- Furthermore, the definition of 'microbe' (and the adjectival form 'microbial'), and the usage conventions of 'algae' and 'microalgae' are now provided under the 'phylogeny and terminology' section of the supplementary materials.

The reference to algal (and prokaryotic, e.g., Siphonophycus spp.) assemblages from the 1Gyr-old Torridon Gp is welcome but the authors should also insert a reference to other Meso-/Neoproterozoique assemblages which are taxonomically better constrained (e.g. Roper Gp, Australia). It has to be highlighted that virtually any pre-Ediacaran palynological assemblage is dominated by or include large proportions of leiospheres.

- We agree, but the Torridon Group was chosen for comparison because these microfossils represent continental microfossil assemblages. A broader comparison with marine successions (e.g., Roper Group) would definitely be interesting, but we argue would be beyond the scope of the present study.
- Of even greater relevance is the fossil assemblage of the Nonesuch Formation, USA. A very recent review of this assemblage offers greater insights into the lacustrine microbial communities of the Mesoproterozoic (Strother & Wellman, 2021, *J. Geol. Soc.* 178: jgs2020-133). We argue that this assemblage is of greater relevance to the present paper because of its: 1, better age control; 2, more diverse assemblages (particularly among the cyanobacterial remains); 3, similar preservation of organic remains to the microbial fossils of the present study; and 4, recent review, which has included up-to-date findings and interpretations. Hence, we have opted to substitute the reference to the Torridon Group assemblage with that of Nonesuch Formation assemblage. Since the assemblages are very similar and mutually supportive, the remaining text of this sentence can largely stay the same.

Reviewer #3 (Remarks to the Author):

Researchers studying mass extinctions often focus on the taxa that go extinct, while the surviving taxa can actually provide information on what happens with ecosystems during the survival and recovery interval. And this is exactly what Mays et al. do, focusing on the unusual long-term proliferation of algae in the Sydney Basin, comparing these patterns with other areas in southeastern Gondwana, and similar signals during other mass extinction events.

P 2, line 35. “extreme global warming (c. 6–12°C increase in the tropics)”. Just by looking at the current warming of the boreal regions, a potentially threefold increase in CO₂ levels must have raised the temperatures and changed the rainfall regime in the higher latitudes immensely. There have been other papers that modeled the temperature increase as a result of the CO₂ outgassing of the Siberian Traps; please expand the number of sources for climate reconstruction. When doing so, could the authors give us an idea of how gradual the temperature increase would have been, given that the traps have been active over quite a long period, and what could have caused the system during that long period of Traps emissions to reach a threshold (the EPE) and trigger sudden biotic change over a large geographic region? Also, what would have been the magnitude estimated temperature increase for the high latitude regions at the end of the Permian?

- Estimated temperature increases for the high southern latitudes have now been included in the manuscript, supported by an additional recent paper (led by TF, and co-authored by CF, VV, SM and CM; *Geology*, v. 49). The findings of this paper are directly relevant to the present manuscript, since they are based on new, proxy-based estimates from the Sydney and Bowen basins, eastern Australia.
- An additional reference to the latest estimates of CO₂ increase for the end-Permian event has also been provided (Wu et al. 2021, *Nature Communications* 12: 2137). We have replaced the existing reference for CO₂ estimates, since the more recent paper builds upon the findings of the former (and both came from the same research group). This latest paper also provides an estimate on the timeframe; this has now been provided in the manuscript.

- Both of the sources above support the climatic conditions that would have promoted the proliferation of algal and bacterial blooms in post-EPE continental settings.
- We agree that the cause of this spike in CO₂ and how this relates to Siberian Traps magmatism is a very interesting avenue. For instance, in a paper led by CF (and co-authored by TF, VV, SM and CM; *Nature Communications* 10, 385), the timing of the glossopterid biome collapse is linked to the onset of the primary extrusion phase of the Siberian Traps. However, we would argue that these finer details are beyond the scope of the present paper.

P 2, lines 36-38. "The most consequential long-term changes on land included the abrupt demise of wetland glossopterid forests of the temperate Southern Hemisphere 8, 9 and the tropical coal-forming forests of east Asia". The text here seems to indicate that members of the glossopterid lineage disappeared during the EPE and based on the terminating *Glossopteris* icon in Fig 2. and earlier papers by the same team that is the case. However, in the same figure 2, the black shaded areas seem to indicate pollen produced by glossopterids and the palynomorph category continues to be present until the very top of the figure (Spatian). If the black areas represent similar pollen types, but produced by different parent plants than glossopterids, that has to be clarified in the text and figure caps. If the *Glossopteris* pollen decline but linger on, could the authors then indicate when this group roughly disappears from the Sydney Basin?

- Yes, *Glossopteris* disappears at the EPE based on current fossil macrofloral data. The persistence of pollen similar to that of glossopterids into overlying strata is the result of two problems: (1) that glossopterid-type pollen (*Protohaploxylinus* and *Striatopodocarpites*) were also produced by some other gymnosperms (e.g., peltasperms: see Balme 1995); and (2) reworking of pollen in a dominantly fluvial succession inevitably extends the apparent range of such pollen types. To avoid the ambiguity, we have included a short explanation in the text and caption to clarify this, and re-labelled the leaves in Fig. 2 ('*Glossopteris*' rather than 'glossopterid').

The glossopterids, which formed dominant vegetation in southern Gondwana, disappear when the concentration of algal remains goes through the roof. These two events almost certainly were brought about by the same ultimate cause, the periodic eruption of the Siberian Traps. Based on conditions during extant algal bloom phenomena, the authors argue that climate change caused the decline of glossopteris, which lead to deforestation, soil erosion and eutrophication of fresh and shallow marine water bodies. This sounds like a plausible explanation for what happens at the EPE, but it is not clear to me how this would have inhibited the recovery of freshwater ecosystems for hundreds of millennia. Palynological records show that other groups (terrestrial plants) establish themselves rapidly in the lowlands, including the more drought and high temperature tolerant seed ferns *Dicroidium* and *Lepidopteris*, and at a later stage the conifer *Voltziopsis*. These taxa that must have been present in the region were clearly better adapted to the new conditions and seemed to have been permanent elements of a relatively stable low diversity recovery flora. Is there evidence that for soil erosion during the Late Post EPE, and were the Siberian Traps still active during this time? If not, what could have been the cause for the algal dominance during that phase?

- Regarding the inhibition of freshwater ecosystem recovery, the essence of this argument for the early stage of freshwater ecosystem stress is outlined in the manuscript as:
 - o "Enhanced weathering intensity and destabilisation of soils following deforestation promoted nutrient influx into the floodbasins. Combined with elevated CO₂, temperature and precipitation seasonality, these factors promoted numerous intermittent pulses of algal and bacterial proliferation."

- To clarify our stance, the manuscript text has been updated to indicate that our data only suggest inhibited recovery of freshwater ecosystem for the first c. 100 kyrs (during the interval of definite microbial blooms).
- However, the above point only addresses the earliest post-EPE interval. Following this interval, the persistence of high algae and bacterial abundances was facilitated by: sustained high CO₂ and temperatures, seasonally high water tables (and the formation of ephemeral lakes), and increased weathering rates which would have promoted a sustained nutrient supply. This argument was presented in the manuscript, but as noted above, we have now clarified that the initial bloom interval was restricted to the first c. 100Kyrs.
- Furthermore, we have included a reference to a recent study that indicates that Siberian traps magmatism persisted intermittently until at least the mid-Smithian (250.60 Ma; Augland et al. 2019, *Scientific Reports* 9: 18723), which would have promoted a favourable climate for flourishing (if not 'blooming') algal and bacterial populations for at least this long. Further, we point out that persistent (long-term) microbial blooms may, themselves, have inhibited freshwater ecosystem recovery by suppressing populations of microbe-grazing organisms via lowering oxygen levels through respiration dominance at night, and via generation of toxic metabolites.

Supplementary Figure 3 and 4. The botanical affinity of the pollen and spores seem in order, but it would be good to provide information on the sources on which they are based. That way the readers don't have to go looking for this information.

- These have now been included in the captions of Supplementary Figures 3 & 4.

Reviewers' Comments:

Reviewer #2:

Remarks to the Author:

The article 'Lethal microbial blooms delayed freshwater ecosystem recovery following the end-Permian extinction', in its updated version, represents an outstanding contribution to the knowledge of past freshwater ecosystems as the conclusions reached by the authors for the end-Permian event could be transferred to other locations and intervals. The conclusions and paleoenvironmental inferences are built up on material with an exceptional quality and wide spatial coverage and based on the use of original complementary techniques. From a methodological point of view, the refined use of paleobotanical, palynological and geochemical data to achieve a robust ecological framework will also serve as a reference to study other upper deltaic settings. I would then recommend strongly the publication of the article in its present state

Reviewer #3:

Remarks to the Author:

The authors have addressed my questions and I would be happy for the paper to proceed.

Comments to the Editorial Team

Please see our responses to the requests of the Editorial Team in the 'Extended Response' document. We have also revised the Reporting Summary and Editorial Policy Checklists, and uploaded these separately.

Reviewer comments (in black) and author responses (in blue)

Reviewer #2 (Remarks to the Author):

The article 'Lethal microbial blooms delayed freshwater ecosystem recovery following the end-Permian extinction', in its updated version, represents an outstanding contribution to the knowledge of past freshwater ecosystems as the conclusions reached by the authors for the end-Permian event could be transferred to other locations and intervals. The conclusions and paleoenvironmental inferences are built up on material with an exceptional quality and wide spatial coverage and based on the use of original complementary techniques. From a methodological point of view, the refined use of paleobotanical, palynological and geochemical data to achieve a robust ecological framework will also serve as a reference to study other upper deltaic settings. I would then recommend strongly the publication of the article in its present state

- No changes required.

Reviewer #3 (Remarks to the Author):

The authors have addressed my questions and I would be happy for the paper to proceed.

- No changes required.